# Betaine and L-Carnitine Synergistically Influence the Metabolome and Immune Response in Dogs

**DOI:** 10.3390/ani14030357

**Published:** 2024-01-23

**Authors:** Dennis E. Jewell, Selena K. Tavener, Renea Creech, Kiran S. Panickar

**Affiliations:** 1Department of Grain Science and Industry, Kansas State University, Manhattan, KS 66506, USA; 2Hill’s Pet Nutrition Inc., Overland Park, KS 66211, USAkiran_panickar@hillspet.com (K.S.P.)

**Keywords:** cytokines, canine, immune cell stimulation, metabolomics, inflammation

## Abstract

**Simple Summary:**

Thirty-two dogs were fed one of four foods for 90 days. These foods contained either no added betaine or L-carnitine, added betaine, added L-carnitine, or both added betaine and L-carnitine. Initially and at the end of the feeding period, immune cell activity and circulating metabolite concentrations were measured. Dietary betaine and L-carnitine both changed the metabolomics milieu, while the combination significantly increased immune cell cytokine release when stimulated with LPS while decreasing cytokine release in the absence of stimulation. This change in cytokine release reflects a healthy reduction in activity associated with basal inflammation, coupled with a strong response to bacterial cell introduction.

**Abstract:**

This study used thirty-two dogs, which were assigned to a preferred period of 14 days and then assigned to one of the four treatment foods: control (containing no added betaine, no added L-carnitine), control with 0.5% added betaine (Treatment 2), control with no added betaine and 300 ppm added L-carnitine (Treatment 3), or control with 0.5% added betaine and 300 ppm added L-carnitine (Treatment 4). All treatment foods were fed for ninety days. Untargeted blood metabolomic analysis and immune response were measured at the beginning and end of the 90-day feeding trial. Feeding betaine increased single-carbon metabolites while decreasing many carnitine-containing metabolites. Feeding L-carnitine increased many carnitine metabolites, while the combination synergistically influenced the metabolome. The combination of betaine and L-carnitine increased the cytokines released in a Tru-culture system in response to stimulation while numerically decreasing their release when unstimulated. Therefore, the combination of dietary betaine and L-carnitine could have the dual positive effects of reducing cytokine stimulation, controlling inflammation during health, and providing a robust response to bacterial infection.

## 1. Introduction

Dietary betaine has the ability to function through both being a methyl donor and thereby influencing single-carbon metabolism and through being an osmolyte, affecting health through direct osmotic influence. In its role as a methyl donor in single-carbon metabolism, betaine donates a carbon to homocysteine, creating methionine. Methionine can then act as a methyl donor through S-adenosyl methionine (SAM). Betaine also acts on immune modulation through NF-κB and its downstream-regulated cytokines [1]. This pathway controls a number of cytokines specifically involved as pro-inflammatory cytokines, such as tumor necrosis factor alpha (TNF-α), interleukin 1 beta (IL-1β), and interleukin 23 (IL-23) with a general reduction in inflammatory cytokines. This has been reported as a response to increased dietary intake not only through NF-κB but also through the NOD-like receptor (NLRP3) and caspase-8/11 [2].

Betaine and L-carnitine were both independently shown to depress the pro-inflammatory cytokines TNF-α, IL-1β, IL-6, and IL-8 in primary human corneal epithelial cells exposed to hyperosmotic stress [3]. Conversely, L-carnitine alone had a stimulatory effect on human peripheral blood cells in young subjects (<30 yr old) but not in all older subjects (70 yr and older) [4]. In human monocytes, L-carnitine reduced TNF-α and IL-12 concentrations in response to LPS stimulation [5]. Dietary L-carnitine has been shown to increase lean body mass and act as an antioxidant in dogs; both could be of value in reducing inflammation, as shown by a reduction in the inflammatory marker enzyme creatine phosphokinase [6]. Carnitine has been reported to have a general benefit as a supportive nutrient for optimum immune function in human disease [7] and recovery [8]. Because both L-carnitine and betaine have been independently shown to influence immune responses, this study was designed to evaluate them independently and when combined together when consumed in dog food. The null hypothesis was that there would be no difference associated with treatment. We hypothesized that these dietary ingredients would provide immune system support that would aid in the management of health and disease.

## 2. Materials and Methods

### 2.1. Pets

This study was approved by the Institutional Animal Care and Use Committee, Hill’s Pet Nutrition, Inc., Topeka, KS (Approval # CP829). There were 32 dogs used in this study (Table 1, 7 beagles, and 1 mixed-breed dog in each of the four treatments).

All pets were healthy based on annual physical examinations and normal blood concentrations of selected analytes. All dogs were fed the control food for 14 days before the start of this study and then fed one of the treatment foods for the following 90 days. Initial (day 0) and final (day 90) blood samples were analyzed for metabolomics and immune response.

### 2.2. Foods

The base food was formulated as a complete and balanced adult dog food without added betaine or a specific source of L-carnitine. The base food was then modified either with added L-carnitine, betaine, or both. The analysis of these foods is reported in Table 2. Each pet was assigned to one food for the duration of the study.

### 2.3. Metabolomics

Plasma samples were collected at the beginning and end of the study and stored at −80 °C until analyzed. A commercial laboratory (Metabolon, Morrisville, NC, USA) performed the untargeted metabolomic analysis as described previously [10].

### 2.4. TruCulture

The TruCulture^®^ assay (Rules Based Medicine, Austin, TX, USA) was conducted following the manufacturer’s instructions for use with minor modifications. TruCulture tubes contain cell culture media +/− immune stimulant, allowing for an ex vivo assessment of immune function. Whole blood was first collected into TruCulture tubes containing either lipopolysaccharide (LPS) or in an unstimulated TruCulture tube (Null/Control) from canines using a Sarstedt Sterile Multifly^®^ Needle Set (19G-21G) and a Sarstedt Monovette^®^ Priming Tube (Rules Based Medicine, Austin, TX, USA). Each tube was pre-weighed before collection, and a post-weight was recorded immediately after collection. The percent difference (in mL) from each 1 mL blood draw was within ±15%. Samples were immediately placed upright in a pre-heated dry heatblock and incubated at 38.6 °C for 24 h. Following incubation, a valve separator (Rules Based Medicine, Austin, TX, USA) was inserted, and the supernatant was collected. Supernatants were stored at −80 °C until further analysis.

### 2.5. Cytokine Analysis

Cytokine analysis was completed on the supernatant of the incubated cells as described above. Analyses were performed using the Canine Cytokine/Chemokine Magnetic Bead Panel ELISA kit (CCYTOMAG-90K, Millipore-Sigma, Burlington, MA, USA) following the manufacturer’s directions. Analysis was performed for the following cytokines: interleukin-6, interleukin-7, interleukin-8, interleukin-10, interleukin-15, TNF–alpha, and MCP-1. Briefly, 25 µL of sample, assay buffer, and antibody-immobilized beads were incubated overnight in a 96-well plate on a shaker at 4 °C. The beads were then washed twice with 200 µL of wash buffer and incubated with 25 µL detection antibodies for 1 h on a shaker at room temperature. Subsequently, 25 µL of streptavidin-phycoerythrin was added to each well and incubated for 30 min on a shaker at room temperature. The beads were then washed twice with 200 µL wash buffer, and 100 µL sheath fluid (40-50,021, Millipore-Sigma, Burlington, MA, USA) was added to each well and placed on the plate shaker for 5 min at room temperature. Fluorescence was detected on the Luminex^®^ 200™ Instrument (Luminex Corp., Austin, TX, USA), and the data were analyzed using the Milliplex^®^ Analyst Software v5.1 (Millipore-Sigma, Burlington, MA, USA). All samples were run in duplicate, and the sum of the molar concentrations of these cytokines was used as the response variable.

### 2.6. Statistical Analysis

Statistical analysis was completed using SAS 9.4 (SAS Institute, Cary, NC, USA). To test the hypothesis that there was an effect on immune response, betaine, carnitine, time, and phase (stimulated with LPS or not stimulated) were used as the fixed variables, with each pet being the experimental unit. Proc mixed was used for analysis. For repeated measure analysis, pet ID was used as a random variable. A *p*-value < 0.05 was used as a significant cutoff. The immune cell value used as a response variable was the natural log of the molar concentration of the total cytokines measured. Statistical analysis through the analysis of variance of the metabolomic data were carried out after the imputation of any missing value, which was assumed to be at the limit of detection for that analyte (minimum value imputation). The statistical analysis was performed on a natural log-transformed data. An analysis of the variance of the concentration at the end of the feeding period was performed. Metabolomic analytes with a concentration difference at the end of the feeding period as compared to the controls with *p* < 0.05 and *q* < 0.1 were concluded to be changed by treatment.

## 3. Results

### 3.1. Blood Metabolomics

When comparing the changes in biochemical metabolites between the groups, there were 723 plasma metabolites measured. There were 71 metabolites that were changed (*p* < 0.05, *q* < 0.1) in the treatment groups as compared to the controls (Table 3). Betaine inclusion in the food resulted in more changes either alone (84 changed), while changes associated with carnitine (63 changed) intake were intermediate as compared to the combination of betaine and carnitine (26 changed) when compared to the dogs eating the control food. Betaine supplementation resulted in an increased concentration of many single carbon metabolites such as sarcosine, dimethyl glycine, methionine, and methionine moieties (methionine, N-acetylmethionine, methionine sulfoxide, N-acetylmethionine sulfoxide, and S-adenosylhomocysteine). Dietary betaine reduced the circulating concentration of a number of carnitine-containing compounds, while dietary carnitine resulted in an increased concentration of many of these compounds, as shown in Table 3. The combination of dietary betaine and carnitine resulted in the maintenance of the increased concentration of most of the single carbon metabolites and no change (in comparison to the controls) in the concentrations of most of the carnitine-containing analytes. There was an increase in taurine concentration in all of the treatment groups.

### 3.2. Immune Response

There was a significant (*p* < 0.01) interaction between betaine, carnitine, time on treatment (initial vs. final), and LPS stimulation of the immune cells. This significant interaction is shown by the decreased cytokine concentration in the unstimulated cells in the dogs consuming increased betaine and carnitine as compared to the enhanced response to stimulation in that group as compared to no changes in the dogs eating the unsupplemented control food (Table 4). This interaction shows that canine immune cells respond differently to either non-stimulation or stimulation by LPS when the dogs are fed the combined betaine + carnitine food, as compared to the response after consuming either the control food or food containing added betaine or carnitine independently. The individual cytokine data are reported in Appendix A.

## 4. Discussion

Betaine serves two principal physiological roles: functioning as a methyl donor for the synthesis of the proteinogenic amino acid methionine and as an osmolyte capable of facilitating water retention in cells to protect cells against osmotic stress. Following a methyl group donation to homocysteine to form methionine, betaine is converted to dimethylglycine, which can then be further metabolized to sarcosine. Here, elevated systemic levels of betaine, methionine, dimethylglycine, and sarcosine in both betaine-supplemented groups (alone or in combination with carnitine) at the end of the study versus the control cohort confirm the efficacy of the addition of this metabolite in influencing the single carbon metabolism pathway.

The increase in circulating methionine concentration via betaine-mediated homocysteine recycling can provide a metabolomic benefit, as methionine plays a pivotal role in multiple biological functions. Being a proteinogenic amino acid, methionine serves as a protein building block as well as being a substrate/precursor for several important downstream metabolites, including the universal methyl donor S-adenosylmethionine (SAM), the antioxidant glutathione, and the osmolyte taurine. Although SAM was not detected here, its post-methyl-donating product, S-adenosylhomocysteine (SAH), is significantly and robustly elevated following betaine supplementation, either alone or in combination with carnitine at the end of the feeding period. This suggests that diet fortification with betaine promotes one-carbon metabolism by efficiently re-methylating homocysteine to methionine, which can then be used to generate additional methyl-donating SAM.

Betaine and carnitine are both known to influence fatty acid metabolism, and this may be part of the subsequent immune system effect. Increased dietary betaine was reported to reduce the adipose size of specific storage depots in geese [11]. Betaine also influences energy use by changing the effects of insulin on porcine adipose cells in explants [12]. In swine, betaine also influenced both the gene expression of uptake proteins and the concentration of fatty acids in muscle, which is associated with increased fatty acid oxidation [13]. In mice’s muscles and livers, dietary betaine consumption resulted in an increase in acylcarnitines [14]. These data reported here are consistent with a conclusion stated in a betaine review “increase in intramuscular carnitine would suggest that betaine may increase carnitine palmitoyltransferase I-mediated free fatty acid translocation into the mitochondria and β-oxidation” [15]. The immune function changes observed in this study, the enhanced activity potential in the presence of LPS stimulation, could then be explained through the change in the energy metabolism of immune cells resulting from the translocation of carnitine into the mitochondria of immune cells, which enhances energy production in response to stimulation.

The enhanced single-carbon pathway may have a number of connections to the immune system. For example, it has previously been reported in dogs [16] that there was an increased concentration of betaine, dimethyl glycine, and sarcosine with the addition of betaine. However, as in this study, when dietary betaine is added alone, there is a decreased concentration of carnitine and a large number of carnitine moieties. This may be a significant component of the enhanced effectiveness seen in the test food with betaine and carnitine combined because, with this combination of dietary betaine and carnitine, there was no reduction in circulating concentration or the large number of carnitine moieties that were decreased by betaine alone. Carnitine can either be made through *de novo* synthesis or be the result of dietary intake. The de novo synthesis pathway requires the amino acids lysine and methionine. Methionine was increased in this study by betaine supplementation. Methionine is then a substrate for S-adenosyl methionine synthesis, which is the methyl donor for lysine, creating trimethyllysine and subsequently carnitine. Therefore, dietary betaine, through the enhancement of methionine and subsequent increase in S-adenosyl methionine, is a support for carnitine synthesis. The increased S-adenosyl methionine, although not measured, may be inferred from the increased concentration of S-adenosyl homocysteine, which is the compound produced from the methylation of lysine in the process of carnitine synthesis. Therefore, it is reasonable to expect that the increased dietary betaine and subsequent increased circulating concentration of betaine would directly increase the production of endogenous carnitine. However, the observed decreased concentration of circulating carnitine and carnitine moieties in response to dietary betaine is therefore somewhat surprising. It is possible that the shift observed in circulating carnitine concentrations with dietary betaine is a shift from circulating carnitine into organ-bound liver and muscle carnitine. This is supported by a carnitine palmitoyltransferase gene expression change that was normalized by dietary betaine in rats with high-fat food consumption [17]. This conclusion is supported by the report that dietary betaine increased acylcarnitines in mice liver and muscle [14]. If dietary betaine similarly caused an internalization of carnitine in this study, it would explain the observed lower carnitine and carnitine moieties in the betaine-only group and the comparatively lowered carnitine moieties in the betaine + carnitine group (as compared to the dogs consuming food with carnitine supplementation alone). These changes in carnitine could be an immune benefit through increased lipid metabolism and enhanced energy availability for immune function when the cells were stimulated with LPS.

The increased methionine and taurine seen in the presence of dietary betaine and carnitine may have been influential in the observed immune response. The role of taurine and methionine in the immune response in dogs has been reviewed [18]. In this review, it was noted that the role of taurine is not well elucidated in lymphocytes. However, it is present at a high concentration in immune cells, and because taurine chloramine decreases TNF-alpha and other immune stimulators, it was pointed out that there is likely a therapeutic benefit to taurine in helping to control an acute inflammatory response. This aid in the management of an acute inflammatory response was observed and may be partially the result of the increased circulating taurine concentration. Methionine has a significant role in controlling the immune response. The role of methionine in influencing the immune system has been reviewed [19]. In this review, it was stated that methionine influences the immune response by being a precursor for carnitine, creatine, cysteine, and succinyl-CoA. In this study, methionine was also enhanced in the dogs consuming foods with enhanced betaine, which resulted in an increased concentration of s-adenosyl homocysteine (SAH). The inflammatory response in human macrophages was shown to be mediated by SAH [20]. These changes in methionine and taurine in response to dietary betaine and carnitine may then have been the means by which the dietary changes influenced the immune response.

There was an increased concentration of asparagine in the dogs consuming the effective carnitine + betaine food. Asparagine and its influence on the immune system have been reviewed [21]. In this review, it was noted that asparagine could modulate immune function through direct interaction with lymphocytes. This activity of asparagine was conducted through a number of pathways. Asparagine synthase activity was increased in immune cells (lymphocytes and macrophages) when stimulated (e.g., with mitogens) in the mouse [22]. It was also reported that an increased intracellular asparagine concentration increased ornithine decarboxylase activity [23]. This enhanced ornithine decarboxylase activity in turn enhances polyamine synthesis, resulting in an enhanced immune response [24].

Serine was increased in the dogs consuming the effective betaine + carnitine food, which could also be one of the means by which the food modulated the immune system. Serine has been shown to increase immune responses when enhanced in the culture medium of hybridoma cells [25]. Also, a food limited in amino acids, including serine, was shown to reduce the immune response in chickens [26].

Choline was increased in the dogs with enhanced dietary betaine. Choline influences the immune response, which may be through primary or secondary effects [27]. These authors concluded that there was evidence of an anti-inflammatory effect of choline on immune cells while there was an increased rate of proliferation. This suggests that one of the factors influencing the observed immune response was the change in circulating choline concentration.

The reduction of cytokine release in the absence of LPS stimulation could be beneficial in a number of disease states. For example, we have reported that there is an association of pro-inflammatory cytokines with a part of the renal disease process in dogs [28]. And there are higher pro-inflammatory cytokines in dogs with age or mitral valve disease [29,30], obesity [31], and IBD [32]. In contrast, immunosenescence, characterized by a progressive decline in immune response with age, has been reported in dogs [33] and is associated with the incidence of age-related diseases in humans [34] and dogs [35,36,37]. There is a reduced number of regulatory T cells (Tregs) in the duodenal villi in general in healthy dogs as compared to dogs with IBD [38]. In this study, there is evidence of anti-inflammatory responses to additional choline in innate immune cells, which may be part of the control system influencing the immune response. In dogs with inflammatory bowel disease (IBD), when compared to healthy controls, there is a defect in Treg homeostasis leading to a decline in oral tolerance. It has been observed that there was a general increase in the inflammatory response and an increase in markers of oxidative damage [38]. After obtaining blood samples from three age groups of beagles (<4 yr; 4–8 yr; 8+ yr), there was a reported decrease in the number of CD4+ CD45RA+ T-cells and CD8+ CD45RA+ T cells with age [38]. More recently, a lower proliferative capacity of CD8+ T cells in older dogs compared to younger dogs was reported [39]. It was shown in dogs that CD8+ cells play an important role in fighting infections and are active as antitumor agents [40]. These data suggest that further research investigating the effects of these dietary interventions on CD4+ and CD8+ cells and aging should bring valuable insight into optimizing immune responses in dogs.

These studies highlight the age-related changes in immune function in dogs and thus provide an attractive target for nutritional intervention. The combination of L-carnitine and betaine may play an important role in attenuating immunosenescence in dogs.

## 5. Conclusions

The metabolic changes associated with feeding dietary betaine and carnitine have the effect of enhancing immune response (as measured by cytokine release) after stimulation with LPS in comparison to a reduced response in the resting state. This could have a significant benefit for healthy aging by reducing basal immune cell stimulation and inflammation signaling while enhancing the ability to defend against pathogens through a heightened cytokine response.

## Figures and Tables

**Table 1 animals-14-00357-t001:** Pet characteristics for each group. Data are represented as absolute values or as means with standard deviations or ranges.

	Control	Control + Betaine	Control + Carnitine	Control + Betaine and Carnitine
Number of dogs	8	8	8	8
Male Neutered	4	4	4	4
Female Spayed	4	4	4	4
Mean weight kg (std deviation)	10.7 (2.4)	11.2 (1.7)	11.4 (1.8)	10.7 (1.1)
Mean age years (range)	5.5 (2–10)	6.0 (2–10)	5.9 (2–11)	4.9 (2–9)

**Table 2 animals-14-00357-t002:** Nutrient values of test foods (Analyzed values as percentages kcals calculated ^1^).

Analyte	Control	Control + Betaine	Control + L-Carnitine	Control + Betaine + L-Carnitine
Moisture	8.24	8.34	8.30	8.04
Protein	35.2	35.1	34.5	34.5
Fat	11.0	11.0	11.1	10.7
Crude Fiber	9.00	9.15	9.20	9.15
Ash	5.69	5.62	5.72	5.67
Calcium	0.86	0.88	0.89	0.90
Phosphorus	0.72	0.74	0.72	0.70
Sodium	0.35	0.36	0.36	0.35
C08:0 Octanoic (Caprylic)	0.14	0.18	0.16	0.18
C10:0 Decanoic (Capric)	0.11	0.14	0.12	0.14
C12:0 Dodecanoic (Lauric)	0.80	1.07	0.93	1.07
C14:0 Tetradecanoic (Myristic)	0.35	0.45	0.40	0.45
C16:0 Hexadecanoic (Palmitic)	1.67	1.71	1.69	1.74
C18:0 Octadecanoic (Stearic)	0.50	0.51	0.50	0.52
Sum Saturated Fatty Acids	3.63	4.13	3.86	4.17
Sum Monosaturated Fatty Acids	2.85	2.84	2.85	2.88
Sum Omega 3 Fatty Acids	0.67	0.65	0.65	0.68
Sum Omega 6 Fatty Acids	2.24	2.16	2.25	2.17
Lysine	2.03	2.03	2.07	2.00
Threonine	1.21	1.21	1.25	1.19
Methionine	1.33	1.25	1.26	1.25
Tryptophan	0.31	0.30	0.30	0.29
Betaine mg/kg	388	5860	257	5950
L-Carnitine mg/kg	23	29	374	392
Metabolizable Energy kcal/kg	3341	3382	3417	3399

^1^ Calculated by the canine dry formula previously reported [9].

**Table 3 animals-14-00357-t003:** Effect of betaine and carnitine on metabolite differences ^1^ in dogs.

	Fold Change (Treatment/Control) at End of Study Measurement		
	−Car + Bet	+Car-Bet	+Car + Bet
Biochemical Name	−Car-Bet	−Car-Bet	−Car-Bet
Sarcosine	1.86	1.04	1.86
Dimethylglycine	2.49	1.17	2.51
Betaine	2.2	0.99	2.19
Serine	1.21	1.12	1.34
N-acetylserine	1.54	1	1.37
Alanine	1.3	1.28	1.31
N-acetylalanine	1.27	0.97	1.12
N-methylalanine	2.44	1.14	2.12
N,N-dimethylalanine	0.2	0.88	0.08
Asparagine	1.21	1.21	1.33
alpha-ketoglutaramate	1.81	1.09	1.97
N-acetylglutamate	1.35	1.01	1.16
pyroglutamine	0.78	0.94	0.71
S-1-pyrroline-5-carboxylate	1.94	1.62	1.99
N6-acetyllysine	1.15	1.34	1.15
N,N,N-trimethyl-5-aminovalerate	0.52	0.46	0.27
isovalerylcarnitine (C5)	0.54	1.79	0.98
2-methylbutyrylcarnitine (C5)	0.52	1.07	0.77
isobutyrylcarnitine (C4)	0.57	1.76	1.13
Methionine	7.13	1.23	8.04
N-acetylmethionine	5.94	1.08	4.65
N-formylmethionine	1.5	0.99	1.45
methionine sulfoxide	5.48	1.05	6.71
N-acetylmethionine sulfoxide	8.97	1.4	8.26
S-adenosylhomocysteine (SAH)	3.76	1.12	3.56
5-methylthioribose	2.2	1.55	1.94
2,3-dihydroxy-5-methylthio-4-pentenoate (DMTPA)	7.06	1.46	6.14
S-methylcysteine sulfoxide	1.08	1.25	0.9
Taurine	1.3	1.37	1.4
Proline	1.37	1.23	1.31
N-methylproline	0.56	0.85	0.38
5-methylthioadenosine (MTA)	6.92	1.09	5.86
2-aminobutyrate	1.77	1.12	1.4
2-hydroxybutyrate/2-hydroxyisobutyrate	1.43	0.96	1.13
gamma-glutamylglutamate	0.72	1.04	0.53
gamma-glutamylmethionine	9.55	1.4	9.83
Phenylacetylalanine	2.9	2.54	2.83
succinylcarnitine (C4-DC)	1.42	1.17	1.28
nonadecanedioate (C19-DC)	0.86	0.62	0.95
eicosanedioate (C20-DC)	0.73	0.48	0.77
butyrylcarnitine (C4)	0.69	1.41	0.86
propionylcarnitine (C3)	0.65	1.78	1.22
acetylcarnitine (C2)	0.52	1.19	0.74
stearoylcarnitine (C18)	0.71	0.99	0.76
cerotoylcarnitine (C26)	0.51	1.61	0.84
nervonoylcarnitine (C24:1)	0.44	1.28	0.8
ximenoylcarnitine (C26:1)	0.42	1.74	0.73
arachidonoylcarnitine (C20:4)	0.62	1.25	0.94
dihomo-linolenoylcarnitine (C20:3n3 or 6)	0.71	1.3	0.88
docosatrienoylcarnitine (C22:3)	0.49	1.49	0.83
adrenoylcarnitine (C22:4)	0.67	1.24	1.06
3-hydroxypalmitoylcarnitine	0.53	1.08	0.8
Carnitine	0.66	1.4	1.03
Choline	1.23	1.11	1.16
1-stearoyl-2-arachidonoyl-GPE (18:0/20:4)	0.64	1.21	0.79
1-stearoyl-GPG (18:0)	1.91	1.66	1.45
1-(1-enyl-stearoyl)-2-oleoyl-GPE (P-18:0/18:1)	0.66	0.97	0.78
1-(1-enyl-stearoyl)-2-linoleoyl-GPE (P-18:0/18:2)	0.73	0.9	0.75
palmitoleoyl-linoleoyl-glycerol (16:1/18:2)	2.16	2.05	1.53
oleoyl-linoleoyl-glycerol (18:1/18:2)	1.52	2.09	1.12
linoleoyl-linoleoyl-glycerol (18:2/18:2)	1.66	2.51	1.12
behenoyl dihydrosphingomyelin (d18:0/22:0)	0.81	0.88	0.64
sphingomyelin (d18:0/18:0, d19:0/17:0)	0.74	0.85	0.65
xanthine	1.76	1.66	1.97
5-methyl-2′-deoxycytidine	1.34	0.88	1.19
nicotinamide riboside	1.42	0.81	0.99
1-methylnicotinamide	1.46	0.88	1.16
2-O-methylascorbic acid	1.7	1.08	1.26
pterin	0.91	0.89	0.55
methylnaphthyl sulfate (1)	1.46	1.42	1.73
2-naphthol sulfate	1.48	1.4	1.71

^1^ Values are ratios of differences between treatment groups, with the control group as the denominator. Squares filled with red are above 1 (*p* < 0.05). Squares filled in green were lower than 1 (*p* < 0.05).

**Table 4 animals-14-00357-t004:** The effect of betaine and carnitine on cytokine release in Tru-culture tubes. Values are the LSMeans of change in the natural logs of the sum of the concentrations between initial and final timepoints ± standard errors.

Food	Unstimulated ^1^	Stimulated ^2^	Ratio ^3^
Control	0.00 ± 0.13 ^a^	−0.07 ± 0.09	−0.07 ± 0.14 ^a^
Control + Betaine	−0.15 ± 0.13 ^a,b^	0.09 ± 0.09	0.24 ± 0.14 ^a,b^
Control + Carnitine	0.04 ± 0.13 ^a^	0.04 ± 0.09	0.00 ± 0.14 ^a^
Control + Betaine + Carnitine	−0.31 ± 0.13 ^b^*	0.13 ± 0.09	0.44 ± 0.14 ^b^*

^1^ Unstimulated cells. Effect on unstimulated cells. = ln(Unstimulated final concentration) − ln(Unstimulated initial concentration). ^2^ Stimulated cells. Effect on stimulated cells. = ln(Stimulated final concentration) − ln(stimulated initial concentration). ^3^ Ratio of cytokine changes in cells Ratio. = ln(change in stimulated cells) − ln(change in unstimulated cells). ^a,b^ Means with different superscripts in the same row are different (*p* < 0.05). * Means are different from 0 (*p* < 0.05).

## Data Availability

Data are contained in the article or available from the corresponding author.

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
