# Peer review of "Betaine and L-Carnitine Synergistically Influence the Metabolome and Immune Response in Dogs"

_animals, 2024, doi:10.3390/ani14030357_

Round 1

Reviewer 1 Report

Comments and Suggestions for Authors

This manuscript evaluates the effect of feeding diets to dogs supplemented with betaine, L-carnitine, or both on the plasma metabolome and cytokine ex vivo cytokine secretion of circulating leukocytes. The study is straighforward and well-written and presented. Two minor points:

1) It would be useful to clarify that Metabolon conducted untargeted metabolomics, as that is not clearly stated and my impression.

2) The diversity in cytokine function tested would benefit from reporting of individual cytokine results. Although most of the cytokines included are generally pro-inflammatory, the panel measures cytokines with regulatory function, namely IL-10.

Author Response

  • It would be useful to clarify that Metabolon conducted untargeted metabolomics, as that is not clearly stated and my impression.

Yes, that impression is correct, and we have added the information for clarity.

  • The diversity in cytokine function tested would benefit from reporting of individual cytokine results. Although most of the cytokines included are generally pro-inflammatory, the panel measures cytokines with regulatory function, namely IL-10.

We have added the individual cytokine results and now have them as supplementary tables. 

Reviewer 2 Report

Comments and Suggestions for Authors

The publication submitted for review concerns effects of Betaine and L-carnitine on metabolome and immune response in dogs. It is well-known that the physiological roles of Betaine and L-carnitine in human or farm animals. Our knowledge about affecting factors on dogs is still insufficient.These results could provide scientific evidence for improving immune system in dogs.Unfortunately, I still have found some flaws that require authors' comments.

1.Suggest presenting lines 132-132 in a table format to better identify differences between groups.

2.Correcting clerical error, such as line 95,-80C.

Author Response

1.Suggest presenting lines 132-132 in a table format to better identify differences between groups.

 These data are presented in Table 3.  We have changed the description to highlight where the reader can find them.  We have also expanded the discussion to better highlight the differences. 

2.Correcting clerical error, such as line 95,-80C.

Thank you, we have reread the document and corrected the error.

Reviewer 3 Report

Comments and Suggestions for Authors

I like the paper but the details and data regarding the immunologic findings are insufficient. The discussion of the metabolomic findings was sparse and should have been better. Submit a revision including data that must be in hand and that should require just minor tweaking of the manuscript.

Author Response

I like the paper but the details and data regarding the immunologic findings are insufficient. The discussion of the metabolomic findings was sparse and should have been better. Submit a revision including data that must be in hand and that should require just minor tweaking of the manuscript.

Our response - 

We have greatly expanded the discussion and added the individual cytokine response as supplementary tables.

Round 2

Reviewer 3 Report

Comments and Suggestions for Authors

Your manuscript is much improved and the discussion is now interesting! My only other suggestion for improvement at this point is that the data in the supplement would be much easier to assimilate if presented in graphical form.

Author Response

We had wondered about the best way to provide the supplementary data.  As you requested we've now included it in graphical form.